# ENDO_STAGE Magnetic Resonance Imaging: Classification to Screen Endometriosis

**DOI:** 10.3390/jcm11092443

**Published:** 2022-04-26

**Authors:** Marc Bazot, Emile Daraï, Giuseppe P. Benagiano, Caroline Reinhold, Amelia Favier, Horace Roman, Jacques Donnez, Sofiane Bendifallah

**Affiliations:** 1Department of Radiology, Tenon University Hospital, Assistance Publique des Hôpitaux de Paris (AP-HP), Sorbonne Université, 75020 Paris, France; marc.bazot@aphp.fr; 2Groupe de Recherche Clinique (GRC-6), Centre Expert en Endométriose (C3E), Assistance Publique des Hôpitaux de Paris, Tenon University Hospital, Sorbonne Université, 75020 Paris, France; emile.darai@aphp.fr (E.D.); sofiane.bendifallah@aphp.fr (S.B.); 3Department of Gynecology and Obstetrics, Tenon University Hospital, Assistance Publique des Hôpitaux de Paris (AP-HP), Sorbonne Université, 75020 Paris, France; 4Department of Maternal and Child Health, Gynecology and Urology, Sapienza University, 00161 Rome, Italy; giuseppe.benagiano@uniroma1.it; 5Department of Radiology, Mcgill University Health Center (MUHC) and Co-Director of the Augmented Intelligence Precision Health Laboratory (AIPHL), MUHC Research Institute, Montreal, QC H4A 3J1, Canada; caroline.reinhold@mcgill.ca; 6Centre of Endometriosis, Clinique Tivoli-Ducos, 220, Rue Mandron, 33000 Bordeaux, France; horace.roman@gmail.com; 7Department of Gynecology, Cliniques Universitaires Saint-Luc, Université Catholique de Louvain, 1200 Brussels, Belgium; jacques.donnez@gmail.com

**Keywords:** endometriosis, classification, deep pelvic endometriosis, deep infiltrating endometriosis

## Abstract

**Introduction:** Transvaginal sonography is the first-line imaging technique to diagnose endometriosis, but magnetic resonance imaging is more accurate in staging the extent of lesions, especially for deep pelvic endometriosis. The revised American Society for Reproductive Medicine and Enzian classifications are commonly used to stage the extent of endometriosis. However, a review underlined their weaknesses in terms of complexity, lack of clinical reproducibility and low correlation with surgical complications and fertility outcomes. Thus, to this day, in clinical practice, there is a lack of consensual, standardized or common nomenclature to stage the extent of endometriosis, posing a worldwide challenge. **Objectives:** The aims of our study were to: (i) develop a new classification (entitled Endo-Stage MRI) based on patterns of endometriosis as observed with magnetic resonance imaging; (ii) compare results with those of the rASRM classification; (iii) estimate the Endo-Stage MRI accuracy to predict the rate of surgical complications; and (iv) propose an Endo-Stage MRI system of triage (low, intermediate, high) that correlates with the risk of surgical complications. The goal is to improve the effectiveness of care pathways and allow for the planning of a multidisciplinary approach when necessary. **Patients and methods:** A single-center observational study using available clinical and imaging data. According to anatomical locations and the extent of endometriotic lesions, a standardized classification comprising six stages of severity (0–5) was designed. **Results:** A total of 751 patients with pelvic endometriosis underwent surgery from January 2013 to December 2018 in a tertiary care university hospital. Their Endo-Stage MRI classification was correlated with: (i) the rate of overall complications (grade I–IV Clavien-Dindo classification, (ii) the rate of major complications (grades III–IV) and (iii) the rate of voiding dysfunction requiring self-catheterization lasting more than one month. According to the Endo-Stage MRI classification, stages 0, 1, 2, 3, 4 and 5 were observed in 26 (3%), 156 (21%), 40 (5%), 22 (3%), 290 (39%) and 217 (29%) patients, respectively. Using the proposed Endo-Stage MRI system as triage, low (stages 0–2), intermediate (stages 3–4) and high-risk (stage 5), complications were observed in 29 (13%), 109 (34.9%) and 103 (47.4%) patients, respectively. In multivariate analysis, the Endo-Stage MRI system of triage was strongly predictive of surgical complications and achieved higher accuracy than the revised American Society for Reproductive Medicine classification (AUC: 0.78 (95% CI, 0.76–0.80) vs. 0.61 (95% CI, 0.58–0.64)). **Conclusion:** Our study proposes a new imaging classification of endometriosis coined *Endo-Stage MRI classification*. The results suggest that when applied to a clinical situation, it may improve care pathways by providing crucial information for identifying intermediate and/or high-risk stages of endometriosis with increased rates of surgical complications. To make this classification applicable, a multicentric validation study is necessary to assess the relevancy and clinical value of the current anatomical MRI classification.

## 1. Introduction

Endometriosis is defined as the presence of endometrium-like tissue outside the uterus [1]. A recent systematic review, including 11 studies, analyzed the prevalence of endometriosis in the general population, which ranged from 0.8% to 28.6%, with an overall estimation of 4.4% [2]. In addition, the pooled estimated prevalence of endometriosis was 33.5% in women who underwent surgery for benign gynecological conditions, 23.8% in infertile women, and 49.7 % in women with chronic pelvic pain [2]. 

Transvaginal sonography is the first-line imaging technique to diagnose endometriosis, but magnetic resonance imaging (MRI) is more accurate in staging the extent of lesions, especially for deep pelvic endometriosis (DPE) [3,4,5]. The revised American Society for Reproductive Medicine (rASRM) and Enzian classifications are commonly used to stage the extent of endometriosis [6,7]. However, a review underlined their weaknesses in terms of complexity, lack of clinical reproducibility and low correlation with surgical complications and fertility outcomes [8]. Thus, to this day, in clinical practice, there is a lack of consensual, standardized or common nomenclature to stage the extent of endometriosis, posing a worldwide challenge.

Recently, the World Endometriosis Society (WES) highlighted the need for a reproducible preoperative imaging system of triage to better characterize the extent of endometriosis and improve clinical management [9]. A new classification entitled “Deep pelvic endometriosis classification index” (dPEI) was recently published underlining the value of MRI to stage endometriosis, but with some limitations on the strict definition of the various compartments and a lack of external validation [10].

In line with these recent developments, we carried out an investigation aimed at: (i) developing a new classification system (entitled Endo-Stage MRI) based on patterns of endometriosis on MRI, (ii) comparing its value to the rASRM classification, (iii) estimating the Endo-Stage MRI accuracy in predicting surgical outcomes in terms of complications and (iv) proposing an Endo-Stage MRI system of triage (low, intermediate, high) that correlates with the risk of surgical complications.

The overall intent is to improve the effectiveness of care pathways and allow for the planning of a multidisciplinary approach when necessary. 

## 2. Materials and Methods

### 2.1. Population

The database of the Pathology Department of the Tenon University Hospital, Sorbonne University, Paris, was screened from January 2013 to December 2018 to identify women who had undergone surgery for suspected pelvic endometriosis (*n* = 1293). The database of the Radiology Department was then searched to identify among these patients those who had MRI evaluation before surgery. Patients with preoperative MRI performed outside this institution (*n* = 542) were excluded from the present investigation. 

The study was approved by the Ethics Committee of the National College of French Obstetricians and Gynecologists (CNGOF) (reference number: CEROG 2012-GYN-10-03).

### 2.2. MRI Technique

MRI sequences were acquired at 1.5 T (GE HDXT, Milwaukee, WI, USA) or 3 T (GE Architect, Milwaukee, WI, USA) using a phased pelvic array. The acquisition protocol, including sequences and parameters, followed guidelines recently published by the European Society of Urogenital Radiology (ESUR) (Table 1) [11]. Bowel preparation by enema and antiperistaltic drug administration (Glucagen^®^, Novo Nordsik) were routinely offered to the patients. Vaginal and rectal opacification and gadolinium injection were not included in our routine MRI protocol. 

### 2.3. MRI Evaluation

All MRI examinations were performed and interpreted prospectively by radiologists, and all the data present in the formal reports were entered in an Excel spreadsheet. No attempt was made to review the MRI studies retrospectively to evaluate intra and inter-observer agreement among radiologists. Pelvic endometriosis was diagnosed in accordance with previously described criteria [12,13,14]. 

### 2.4. Surgical and Pathological Findings 

Surgery was performed by different experienced gynecological surgeons according to surgical procedures previously published [15,16,17,18,19,20]. Deep pelvic endometriosis, also called deep infiltrating endometriosis, is defined as infiltration of the implant of endometriosis under the surface of the peritoneum, as previously described [13]. All surgical, pathological and outcomes findings documented in the official medical files were entered in an Excel spreadsheet.

## 3. The Endo-Stage Mri Classification 

Based on the International Federation of Gynecology and Obstetrics (FIGO) classification for staging gynecological cancers and on the British Royal College of Obstetricians and Gynecologists (RCOG) scoring of surgical complexity [21,22], a new standardized classification (coined Endo-Stage MRI), was designed under the supervision of a gynecological subspecialty radiologist (MB) with more than 25 years of experience in MRI of endometriosis and a skilled surgeon (SB) with 5 years of surgical experience in endometriosis. 

Briefly, as shown in Table 2**,** six Endo-stages of MRI (0 to V) were created according to the anatomical location, the extent of endometriotic lesions and European Society of Human Reproduction and Embryology (ESHRE) guidelines [9]. The aim was to follow the evolution of pelvic endometriosis, as suggested by Nisolle and Donnez [23].

**Stage 0**: refers to the presence of only superficial peritoneal endometriosis irrespective of the locations involved (ovarian fossa, vesico-uterine fold, pouch of Douglas) and/or uni or bilateral ovarian endometrial cyst (size ≥1 cm) (Figure 1). 

**Stage I**: defines the presence of retro-cervical DPE, including isolated involvement of the torus (1A), or the torus and uni or bilateral uterosacral ligament endometriosis (1B) (Figure 2). 

**Stage II**: relates to DPE involving the vagina (2A), or the rectovaginal septum (2B) (Figure 3). 

**Stage III**: refers to parametrial (Stage 3A), sacro-recto-genital septum (3B) or lateral pelvic wall (Stage 3C) disease (Figure 4). 

**Stage IV**: indicates the involvement of the bladder (5A) or the rectum (5B) (Figure 5). 

**Stage V**: defines the presence of more than three pelvic DPE locations ≥stage 2 (5A) or more than three pelvic DPE locations ≥ stage 2, associated with the involvement of distant intra-abdominal organs (ileum/cecum/ appendix or diaphragm) (5B) (Figure 6).

Finally, as indicated in the last column of Table 2, an MRI system of triage to predict complications is proposed. This aims at predicting the outcomes of surgery with accuracy in terms of intra or post-operative complications. It is subdivided into three risk categories: *low*, *intermediate* and *high*.

## 4. Statistical Analysis

### 4.1. End Points

The accuracy of the Endo-MRI classification was estimated according to its correlation with: (i) the rate of overall complications (grade I–IV Clavien–Dindo classification (CDC)) [24], (ii) the rate of severe complications (grades II–IV CDC) and (iii) the rate of voiding dysfunction requiring self-catheterization lasting more than one month. 

### 4.2. Predictive Value

The performance of the Endo-Stage MRI classification was quantified with respect to discrimination criteria [25,26]. Discrimination (i.e., whether the relative ranking of individual predictions is in the correct order) was quantified using the area under the curve (AUC) of the receiver operating characteristics (ROC) with a confidence interval (CI). The AUC is a summary measure of the ROC that reflects the ability of a test to discriminate the outcomes across all possible levels of positivity. AUC ranges from 0 to 1, and a model is considered to have a poor, fair or good performance if the AUC lies between 0.5 and 0.6, 0.6 and 0.7 or is greater than 0.8, respectively [27].

### 4.3. Comparison of Classifications

Stages of the new Endo-Stage MRI were compared to those of the rASRM classifications according to discrimination criteria quantified by the receiver operating characteristic curve (ROC-AUC) to estimate their accuracy [27,28]. 

Descriptive analysis was based on Student’s *t*-test and the Mann–Whitney test for parametric and nonparametric continuous variables, respectively, and the Chi-square test or Fisher’s exact test, as appropriate, for categorical variables. Values of *p* < 0.05 were considered to denote differences. 

The data were managed with an Excel database (Microsoft, Redmond, WA, USA) and analyzed using R 2.15 software, available online.

## 5. Results

### 5.1. Epidemiological and Surgical Characteristics of the Population

From the surgical, histopathologic and radiological databases of the Tenon University Hospital, we identified 751 patients who underwent preoperative MRI from January 2013 to December 2018.

The surgical and patients characteristics are summarized in Table 3. The median age and body-mass index (BMI) were 33 years (range: 19–60 years) and 22.5 kg/m^2^ (range: 12–42), respectively. Surgery was performed by laparoscopy, laparotomy and after conversion in 87% (651), 10% (76) and 3% (24) of cases, respectively. According to Endo-Stage MRI classification, Stages 0, I, II, III, IV and V were observed in 26 (3%), 156 (21%), 40 (5%), 22 (3%), 290 (39%) and 217 (29%) patients, respectively. 

### 5.2. Complications Rates According to Endo-Stage MRI Classification and System of Triage

Based on the Endo-Stage MRI system of triage in *low* (stages 0-I-II), *intermediate* (stages III–IV) and *high-risk* (stage V) cases, overall complications rates were observed in 29 (13.0%), 109 (34.9%) and 103 (47.4%) patients, respectively. Using the Clavien–Dindo classification, the more serious complications (grades III–IV CDC and self-catheterization >1 month) were present in 32%, 11% and 16% of the patients, respectively.

Utilizing the Endo-Stage MRI classification, the complications rates were positively and significantly correlated with the different stages: The more severe the disease was, the more important were the complications. Details concerning grades III–IV CDC and self-catheterization are provided in Table 4.

Based on the Endo-Stage MRI system of triage, differences were observed in the overall complication rate (*p* < 0.001), grades III–IV CDC (*p* < 0.001) and self-catheterization rate >1 month (*p* < 0.001) (Table 4).

### 5.3. Accuracy

#### 5.3.1. Predictive Value of Endo-Stage MRI Classification 

Table 5 summarizes the results of the uni and multivariate analysis for predicting overall complications and grades III–IV CDC and self-catheterization >1 month. Using a multivariate analysis, independent of age, BMI, history of surgery and surgical approaches, the Endo-Stage MRI system of triage was statistically associated with poor surgical outcomes, i.e., overall complications (<0.001), grades III–IV CDC (<0.001) and self-catheterization >1 month (<0.001).

#### 5.3.2. Endo-Stage MRI and rASRM Classifications

The respective AUC of Endo-Stage MRI and r-ASRM classification for predicting overall complications rates, grades III–IV CDC and self-catheterization >1 month are reported in Figure 7. This indicates that Endo-Stage MRI classification provides higher accuracy than ASRM (AUC: 0.78 (95% CI, 0.76–0.80) vs. 0.61 (95% CI, 0.58–0.64)).

## 6. Comment 

### 6.1. Principal Findings

The proposed Endo-Stage MRI classification suggests a significant correlation between the higher stages of the disease (stages III, IV and V) and surgical complication rates.

Indeed, the highest-risk stage (stage V) was significantly associated with the occurrence of surgical complications (both during and after the intervention) than intermediate (stages III and IV) or low-risk stages (stages I and II). In comparison to rASRM classification, our classification is preoperative, can guide surgery and is more accurate in predicting and stratifying surgical complications.

### 6.2. Results in the Context of What Is Known

To date, only a few imaging classifications have been proposed to stage endometriosis, one with ultrasound and two others using MRI [7,10,29]. The ultrasound-based endometriosis staging system (UBESS) is the only ultrasound classification assigning stages based on the anticipated level of the complexity of the surgical procedure [29]. Patients are classified as UBESS I, II and III, which correlate with three levels of surgical complexity according to the Royal College of Obstetricians and Gynecologists (RCOG) [22]. In accordance with the UBESS, our proposed classification has the potential to facilitate the triage of women with a higher stage of disease [29]. However, the RCOG surgical score does not detail all the intraoperative complexities that may be encountered during surgery and are mainly intended to determine the level of expertise of surgeons [30].

The rASRM classification continues to be the most widely used classification for evaluating pelvic endometriosis; however, it does not clearly take into account the presence of DPE. Hence, the ENZIAN score was recently introduced to supplement the rASRM classification [7]. The ENZIAN classification provides an artificial division of the pelvic cavity into three main compartments for DPE (vertical (A), horizontal (B) and dorsal (C)) [7]. Although this tool was very innovative, a number of limitations of the Enzian classification remain. First, it does not provide an overall evaluation of pelvic endometriosis. When endometriosis is located at the margin between two intersecting compartments, the lesion is assigned to the larger compartment affected by endometriosis, not to both compartments [31]. Second, parametria (or cardinal ligaments) are not clearly defined. Finally, the measurement of the size of the lesions is unclear.

Recently, Thomassin-Naggara et al. proposed an MRI classification entitled the deep pelvic endometriosis index (dPEI) [10]. The aims of this study were to develop a classification including lateral locations and to predict complications after surgery for DPE [10]. However, although the delineation of the various compartments is relatively straightforward in a healthy pelvis, the distortion of the pelvic structures and organs by endometriosis renders the delineation of these compartments more difficult. Hence, this classification, before being more widely implemented, would require external validation. In contrast to the dPEI classification, the proposed classification takes into account the global extent of endometriotic lesions mimicking the goals of the FIGO classification for cervical cancer. 

There is an important unmet need for a clinically relevant MRI endometriosis classification that allows patient stratification for endometriotic health care management. Adamson highlighted the criteria needed in any proposed endometriosis classification for the World Endometriosis Society in 2011 [32]. Our proposed classification has been designed to address all the criteria listed by Adamson. Indeed, the Endo-Stage MRI classification and system of triage (i) is easy to understand for physicians and patients due to simple and standardized descriptions of anatomical locations of endometriosis, (ii) reflects the anatomical progression of the disease, (iii) provides prognostic information in uni and multivariate analysis concerning post-operative complications and (iv) was empirically designed and scientifically (statistically) derived.

Furthermore, in comparison to the widely used rASRM score, our classification has higher accuracy and greater clinical relevance, to predict overall complications, Clavien–Dindo 3–4 complications, and voiding dysfunction. Indeed, the AUC values were 0.78 (95% CI, 0.76–0.80) and 0.61 (95% CI, 0.58–0.64), 0.71 (95% CI, 0.69–0.73) and 0,60 (95% CI, 0.57–0.63), 0,71 (95% CI, 0.69–0.73) and 0.53 (95% CI, 0.51–0.55), respectively. 

### 6.3. Clinical Implications

We anticipate that the proposed Endo-Stage MRI classification will improve clinical care quality for patients with endometriosis by allowing a multidisciplinary management approach, which can include expert image reads in dedicated specialty expert centers. Imaging data made available through the new method will greatly contribute to enabling pre-surgical informed decision-making for both patients and surgeons. Its value lies in assisting radiologists, gynecologists and surgeons in describing the various patterns of endometriotic lesion locations. 

### 6.4. Research Implications

We believe this imaging classification will facilitate uniform reporting between physicians and may facilitate a better selection of patients for personalized treatment defined by a multidisciplinary management team, including expert surgeons and expert radiologists. As it has been demonstrated in Owoeye et al.’s study regarding sport exercise medicine, the absence of context-specific dissemination and implementation strategies to support the uptake of evidence-based interventions leads to poor execution of interventions and is, therefore, associated with suboptimal outcomes and increased health care costs. Quality theory-based research is needed for the successful dissemination and implementation of evidence-based interventions to address practice gaps [33].

### 6.5. Strengths and Limitations

Several limits of the current study merit discussion. First, we cannot exclude an inherent bias linked to its observational nature since all imaging and surgical data were obtained from an experienced endometriosis center. Hence, we had to exclude from the analysis a significant number of cases in which preoperative MRI was not performed in our center. Second, intra and inter-observer variability were not evaluated, and there is a lack of the considerations of adhesions lesions in the different stages. In addition, the ENDO-stage MRI needs to be evaluated from a clinical point of view with a specific correlation between symptoms and quality of life. The following classification has been developed to improve the description, classification and triage based on MRI findings. It would be interesting in the future to assess the classification value based on ultrasound and computed tomography, especially for the most advanced endometriosis lesions. As well, considering the rectovaginal septum endometriosis described as stage IIA in the current MRI classification, its value must be evaluated prospectively according to surgical findings since the native area of the rectovaginal septum is behind the lower 2 to 3 cm of the vagina, and the pouch of Douglas extends to the middle third of the vagina in 93% of women [34]. However, the level of the pouch of Douglas is modified in the presence of deep endometriosis [1]. Finally, our classification was created through a retrospective analysis of cases and not in a prospective fashion. Therefore, a multicentric prospective study is required to validate the potential value of such classification.

## 7. Conclusions

The proposed Endo-MRI classification system has been designed to allow for uniform reporting of different phenotypes of endometriosis. We believe this imaging classification will facilitate uniform reporting between physicians and improve the effectiveness of patient care pathways. In the future, we plan to conduct a multicentric validation study to achieve clinically relevant improvements and consensus on this reporting system.

## Figures and Tables

**Figure 1 jcm-11-02443-f001:**
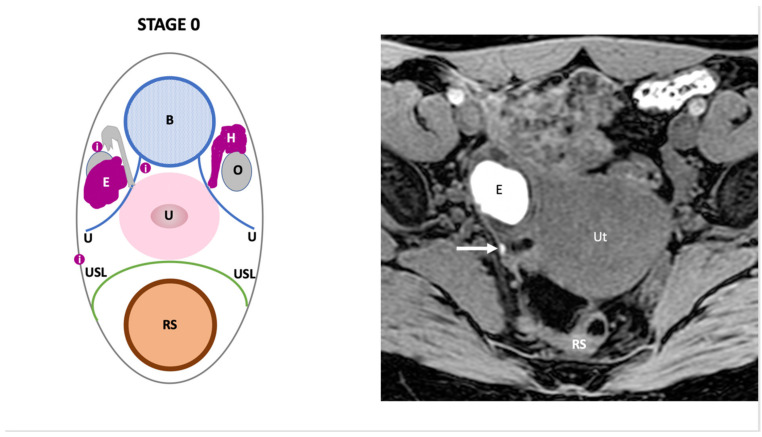
Schematic and example of stage 0 Endo-Stage MRI classification showing the presence of an endometrial cyst (**E**) that displays high signal intensity on a T1-weighted MR with fat-saturation associated with a superficial peritoneal implant (**i**), arrow. Note: (**B**): bladder, (**RS**): rectosigmoid colon, (**O**): ovary, (**U**): ureter, (**H**): hematosalpinx, (**Ut**): uterus, red point: uterine artery at crossing level. (**USL**): Uterosacral ligament.

**Figure 2 jcm-11-02443-f002:**
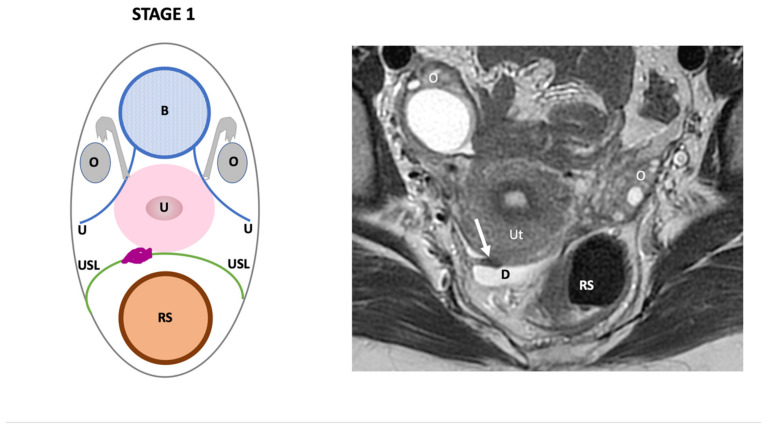
Schematic and example of stage I Endo-Stage MRI classification showing the presence of an irregular thickening of the right uterosacral ligament (arrow) with low signal intensity on a T2-weighted MRI underlined by a small amount of fluid in the pouch of Douglas (**D**). Note: (**B**): bladder, (**RS**): rectosigmoid colon, (**O**): ovary, (**U**): ureter, (**USL**): uterosacral ligament, (**Ut**): uterus; red point: uterine artery.

**Figure 3 jcm-11-02443-f003:**
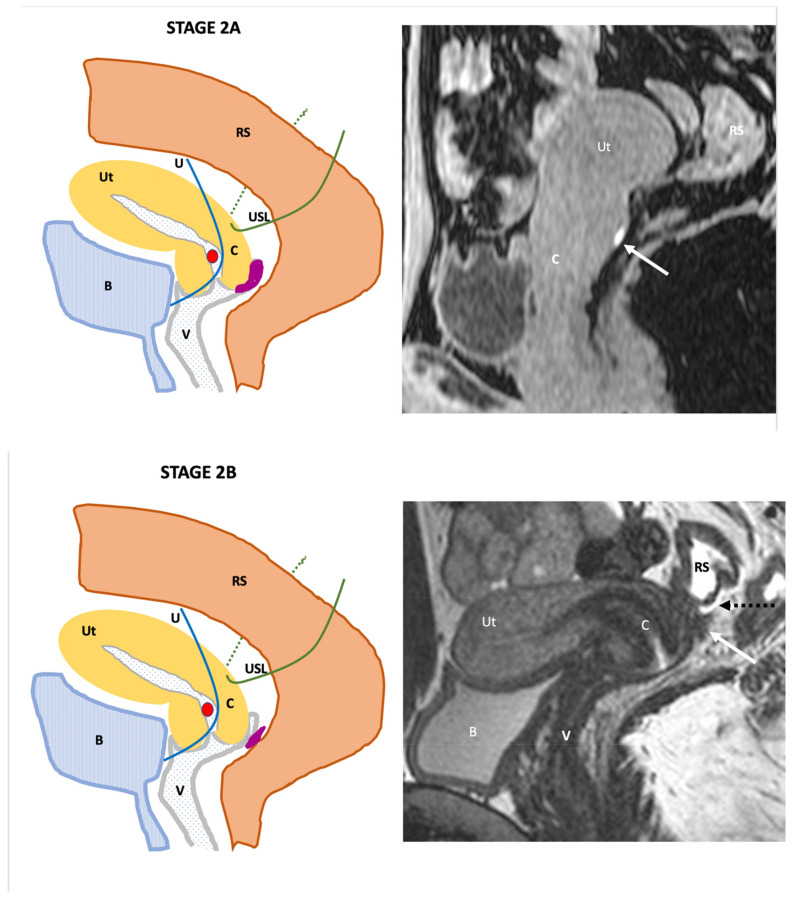
Schematic and example of stage IIA Endo-Stage MRI classification showing the presence of a localized regular thickening of the posterior vaginal wall (arrow) containing high signal intensity on a T1-weighted MRI (arrow). Schematic and example of stage IIB Endo-Stage MRI classification showing the presence of an irregular thickening of rectovaginal septum (arrow) with low signal intensity on a T2-weighted MRI located just below a small amount of pelvic fluid located in the pouch of Douglas (dotted black arrow). Note: (**B**): bladder, (**C**): cervix, (**RS**): rectosigmoid colon, (**U**): ureter, (**USL**): uterosacral ligament, (**Ut**): uterus; red point: uterine artery, (**V**): vagina.

**Figure 4 jcm-11-02443-f004:**
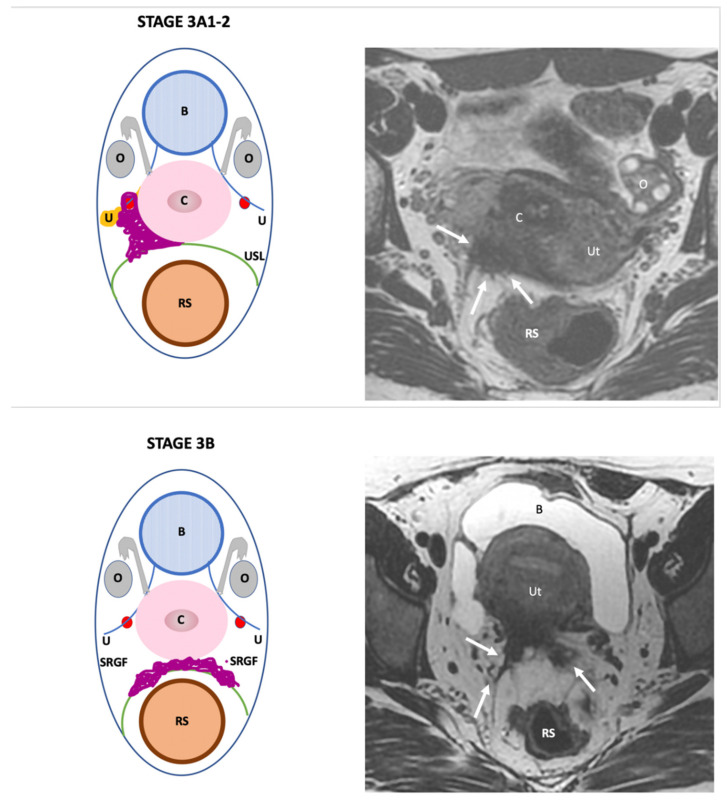
Schematic and example of stage IIIA1 Endo-Stage MRI classification showing the presence of a large irregular mass with low signal intensity on a T2-weighted MRI involving right uterosacral ligament and parametrium (arrows). Schematic and example (of stage IIIB Endo-Stage MRI classification showing the presence of bilateral involvement of sacro-recto-genital fascia that appears irregular and thickened with low signal intensity on a T2-weighted MRI (arrows). Schematic and example of stage IIIC Endo-Stage MRI classification showing the presence of extensive deep posterior and lateral deep endometriosis displaying low signal intensity on a T2-weighted MRI and multiple tiny high signal intensity spots. This lesion involves the vagina (**V**), rectosigmoid colon (**RS**), right sacro-recto-genital fascia and lateral pelvic wall abutting close to piriformis (**P**) and sciatic nerve (dotted arrows). Note: (**B**): bladder, (**C**): cervix, (**RS**): rectosigmoid colon, (**O**): ovary, (**U**): ureter, (**USL**): uterosacral ligament, (**Ut**): uterus; red point: uterine artery, (**SRGF**): sacro-recto-genital fascia.

**Figure 5 jcm-11-02443-f005:**
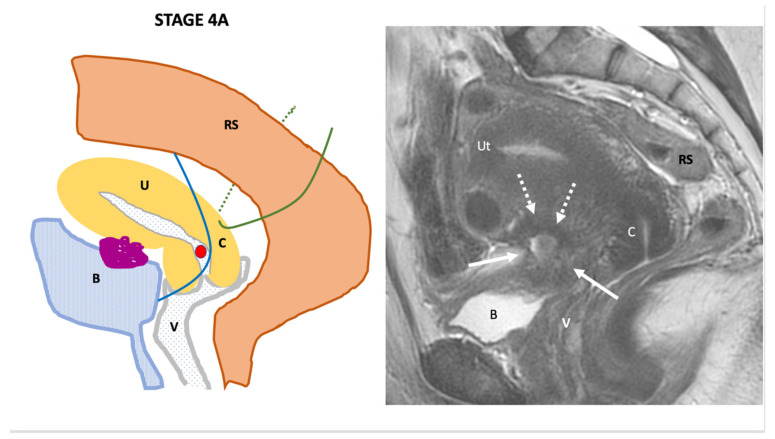
Schematic and example of stage IVA Endo-Stage MRI classification showing the involvement of the vesico-uterine pouch with deep endometriosis involving the bladder (**B**) associated with anterior external adenomyosis (dotted arrows). Schematic (9a) and example (9b) of stage IVB Endo-Stage MRI classification showing the presence of multiple DPE involving the rectosigmoid colon (dotted arrows) and rectum (arrows). Note: (**A**): Ascites, (**B**): bladder, (**C**): cervix, (**RS**): rectosigmoid colon, (**U**): ureter, (**Ut**): uterus; red point: uterine artery, (**V**): vagina. (1,2,3): possible localization of endometriosis lesion.

**Figure 6 jcm-11-02443-f006:**
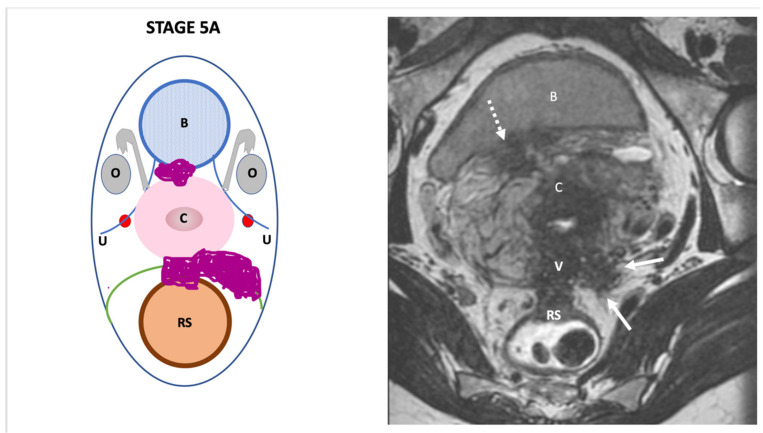
Schematic and example of stage VA Endo-Stage MRI classification showing the presence of more than three deep pelvic endometriotic locations as follows: extensive deep posterior and deep lateral endometriosis displaying low signal intensity on a T2-weighted MRI and multiple tiny high signal intensity foci. This lesion involves the vagina (**V**), rectosigmoid colon (**RS**), left parametrium and sacro-recto-genital fascia (arrows) and bladder (dotted arrow). Schematic and examples of stage VB Endo-Stage MRI classification showing the presence of multiple extrapelvic deep endometriotic locations, some with small high signal intensity foci on a T1-weighted MRI with fat saturation. Note the endometriotic implants located on the undersurface of the right diaphragm (11b, dotted arrow) and on the extrapelvic intestinal sites (11c) showing low signal intensity on T2-weighted implants involving the cecum (arrows) and sigmoid colon (short arrow). Note: (**O**): ovary, (**U**): ureter, (**C**): cecum, (**B**): bladder, (**A**): appendix; (**S**): sigmoid colon, (**SB**): small bowel, (**L**): liver.

**Figure 7 jcm-11-02443-f007:**
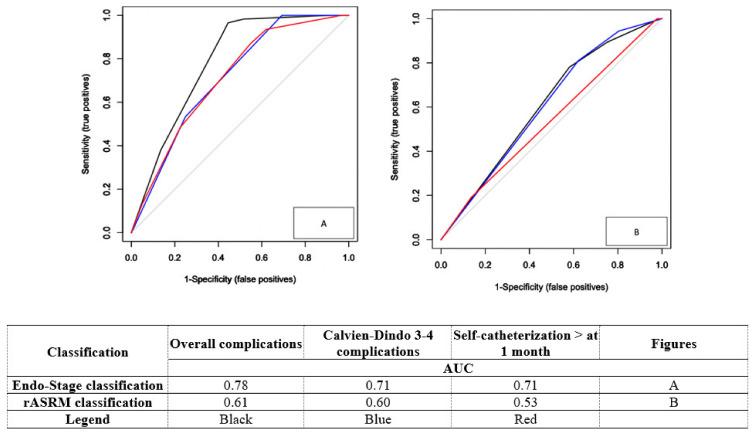
Comparison of Endo-Stage MRI classification and ASRM accuracies to predict surgical outcomes compared to equivalent stages. AUC: Area under the curve, rASRM: The revised American Society for Reproductive Medicine.

**Table 1 jcm-11-02443-t001:** Parameters of MRI sequences.

	1.5 T					3 T				
	Sagittal 2DT2W	Axial 2DT2W	3DT2W	Axial T1W	Oblique Axial 2DT2W	Sagittal 2DT2W	Axial 2DT2W	3DT2W	Axial T1W	Oblique Axial 2DT2W
**TR**	6000	6300	1400	6	4100	10,800	7000	1600	7	7300
**TE**	100	100	110	3	100	110	130	110	1	110
**Angle**	140	160	90	15	160	142	111	90	15	150
**Matrix**	320 × 320	320 × 320	260 × 260	320 × 320	320 × 320	480 × 480	352 × 352	320 × 320	320 × 320	340 × 340
**FOV** **Phase FOV**	24 24	30 30	26 22.1	32 32	24 24	28 28	32 32	28 16.8	32 32	24 24
**Nex**	3.09	1	2	1.4	3.82	1.72	1.54	2	1.21	2.21
**BW (Hz)**	41.7	41.7	41.7	90.9	41.7	83.3	50	62.5	166.7	41.7
**Voxel size (mm)**	0.5	1	0.5	1	0.3	0.5	1	0.5	1	3
**Slice thickness (mm)**	4	5	1	2	3	4	5	1	2	3
**Acquisition time**	3:00	3:20	5:00	2:30	2:30	10,800	7000	1600	7	7300

TR: Repetition Time, TE: Time to Echo, FOV: Field-of-view, BW: Bandwidth.

**Table 2 jcm-11-02443-t002:** Endo-Stage MRI classification.

Endo-Stage MRI	Location	Extent	Risk Stratification
**Endo-Stage MRI** **0**	Superficial endometriosis	Ovarian fossa, vesico uterine fold, pouch of Douglas	**Low**
	Endometriosis is strictly confined to the ovaries	B. Unilateral endometrial cyst C. Bilateral endometrial cysts	
**Endo-Stage MRI** **I**	Endometriosis is strictly confined to the retrocervical area	Torus Torus with uni- or bilateral uterosacral ligament endometriosis	
**Endo-Stage MRI** **II**	Endometriosis invades beyond the retrocervical area, but not to the pelvic side-wall	Vaginal Rectovaginal septum endometriosis	
**Endo-Stage MRI** **III**	Endometriosis extends to the pelvic side-wall and/or causes hydronephrosis or non-functioning kidney	A. Parametrial involvement by DPE A1: without hydronephrosis A2: with hydronephrosis or non-functioning kidney B. Sacro-recto-genital septum involvement C. Lateral pelvic side-wall involvement	**Intermediate**
**Endo-Stage MRI** **IV**	Endometriosis has involved the bladder or rectosigmoid colon	Bladder involvement Rectosigmoid involvement Rectum Rectosigmoid junction Sigmoid colon Multifocal	
**Endo-Stage MRI** **V**	Multiple deep endometrioticlocations	More than **three** DPE locations ≥ stage 2 More than **three** DPE locations ≥ stage 2 with extra-pelvic locations (ileum, cecum, appendix, diaphragm)	**High**

DPE: deep pelvic endometriosis.

**Table 3 jcm-11-02443-t003:** Patients, surgical and MRI characteristics.

**Patient Characteristic (*n* = 751)**		
**Age (years)**	Average ± SD Median/range	33.7 ± 6.55 33/19–60
**BMI (kg/m^2^)**	Average ± SD Median/range	23.3 ± 4.27 22.5/12–42
**Smoking**	Yes	6.4% (48)
**Surgical indication**	Pain	71% (534)
	Pain and/or Infertility	28% (211)
	NA	1% (6)
**Prior pregnancy**	Yes	26.5% (199)
**History of surgery**	Yes	43% (312)
**Surgical characteristics (*n* = 751)**		
**Surgical route**	Laparoscopy Laparotomy Laparo-conversion	87% (651) 10% (76) 3% (24)
**ASRM**	Mean ± SD Median	66.72 ± 43.31 64
Endometrioma surgery	Yes Cystectomy Salpingo-oophorectomy	34.2% (258) 14.4% (108) 16.2% (122)
Salpingectomy	Yes	26.5% (200)
Torus uterinum resection	Yes	78.9% (594)
USL resection	Yes	90% (678)
Partial colpectomy	Yes	26.9% (203)
Hysterectomy	Yes	26.5% (200)
Ureterolysis	Yes	75.4% (568)
Parametrectomy	Yes	37.5% (282)
Ureteral re-implantation	Yes	3.3% (25)
Partial bladder resection	Yes	3.9% (30)
Rectosigmoid colon surgery	Absent Shaving Discoid Segmental resection	19% (142) 14% (104) 16% (119) 51% (386)
Protective stomia	Yes	15.8% (119)
Ileal/cecal resection	Yes	5.4% (41)
Appendectomy	Yes	8.1% (61)
Endo-MRI classification	Endo-MRI stage 0 Endo-MRI stage 1 Endo-MRI stage 2 Endo-MRI stage 3 Endo-MRI stage 4 Endo-MRI stage 5	3% (26) 21% (156) 5% (40) 3% (22) 39% (290) 29% (217)
**Endo-MRI stratification**	Low Intermediate High	30% (222) 41% (312) 29% (217)

**Table 4 jcm-11-02443-t004:** Rates of surgical complications according to Endo-Stage MRI classification and system of triage.

	Frequency	Complications Rates According to ENDO-MRI Classification						
		Low Risk Group (*n* = 222)			Intermediate Risk Group (*n* = 312)		High Risk Group (*n* = 217)	*p*-Value
		Stage 0 (*n* = 26)	Stage 1 (*n* = 156)	Stage 2 (*n* = 40)	Stage 3 (*n* = 22)	Stage 4 (*n* = 290)	Stage 5 (*n* = 217)	
Overall complications rate (*n* = 751)	32% (241)	12% (3)	13% (20)	15% (6)	32% (7)	35% (102)	47% (103)	*p* < 0.001
Clavien-Dindo 3–4 complications (*n* = 751)	11% (82)	0% (0)	3% (5)	8% (3)	9% (2)	10% (30)	19% (42)	*p* < 0.001
Self-catheterization at 1 month (*n* = 579)	16% (92)	15% (3)	5% (7)	17% (6)	13% (2)	13% (28)	31% (46)	*p* < 0.001

**Table 5 jcm-11-02443-t005:** Multivariate analysis of factors predicting the risks of post-operative complications.

Variable		Overall Complications			Clavien-Dindo 3–4 Complications			Self-Catheterization > 1 Month		
		Odds Ratio	95% CI	*p*-Value	Odds Ratio	95 % CI	*p*-Value	Odds Ratio	95 % CI	*p*-Value
**Age**		1.01	0.98–1.03	0.72	1.04	0.99–1.08	0.06	0.99	0.95–1.03	0.47
**BMI**		0.97	0.93–1.01	0.21	0.96	0.90–1.02	0.19	0.98	0.92–1.04	0.43
**History of surgery**		2.05	1.40–2.98	**<0.001**	1.69	1.01–2.84	**0.04**	1.35	0.83–2.21	0.23
**Surgical route**	*Reference =* Laparoscopy Laparotomy Laparo-conversion	2.22 9.37	1.24–3.97 2.51–34.92	**<0.001**	1.65 3.16	0.83–3.27 1.11–9.02	**0.06**	1.20 1.79	0.58–2.46 0.55–5.86	0.59
**Endo-MRI stratification**	*Reference = low-risk* Intermediate High	3.30 6.83	1.97–5.53 3.96–11.78	**<0.001**	2.48 6.39	1.04–5.88 2.73–14.99	**<0.001**	1.48 3.73	0.75–2.92 1.92–7.25	**<0.001**

ENDO_STAGE Magnetic Resonance Imaging MRI, Classification to Screen Endometriosis.

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
