# Peer review of "ENDO_STAGE Magnetic Resonance Imaging: Classification to Screen Endometriosis"

_jcm, 2022, doi:10.3390/jcm11092443_

Round 1

Reviewer 1 Report

Introduction is well written with a good explanation of the hypothesis. The research is well documented with recent references
Material and methods are clear and concise. The inclusion and exclusion criteria of the study are very well established and applied. The new algorithm of endometriosis diagnostics is well sustained.
Results. This section is interesting because the characteristics of the study group are presented in clear tables and also the drawings and MRI figures offer a more specific interpretation. 
Discussion. In this section the results are integrated in the actual literature with the peculiarity that this diagnostics method is new. It is interesting how the authors make reference to other classifications of endometriosis diagnostics.
In conclusion it is perfect that authors are talking about the necessity of evaluation of the method regarding inter-observers differences and reproducibility.

Author Response

Reviewer 1

1

Introduction is well written with a good explanation of the hypothesis. The research is well documented with recent references

Thank you for this comment.

2

Material and methods are clear and concise. The inclusion and exclusion criteria of the study are very well established and applied. The new algorithm of endometriosis diagnostics is well sustained.

Thank you for this comment.

3

Results. This section is interesting because the characteristics of the study group are presented in clear tables and also the drawings and MRI figures offer a more specific interpretation.

Thank you for this comment.

4

Discussion. In this section the results are integrated in the actual literature with the peculiarity that this diagnostics method is new. It is interesting how the authors make reference to other classifications of endometriosis diagnostics.

Thank you for this comment.

5

In conclusion it is perfect that authors are talking about the necessity of evaluation of the method regarding inter-observers differences and reproducibility.

Thank you for this comment.

Reviewer 2 Report

The manuscript ‘ENDO_STAGE Magnetic Resonance Imaging: Classification to 2 Screen Endometriosis’ has been reviewed with interest. The data are nice but the conclusions are not or insufficiently supported by the data.  

The major problems are that deep endometriosis and predictive values are poorly defined

  • The MRI classification is a mixture of volume (stage 0 and I) extent (II  III )  and localisation (IV) of deep endo whereas VI is a poorly defined group including the diaphragm, the ileum, the appendix and the caecum.
  • Since all women were operated on, it would be nice to start with the accuracy of estimated dimensions/presence/extend by MRI in comparison with surgery for each localisation
  • That more extensive lesions correlate with a grading of postoperative complications is a self-fulfilling prophecy and thus little informative.
  • The area under the curve is a poor estimation of positive and negative predictive values. In order to conclude that MRI is useful, an estimate of precision and confidence limits would be needed. 
  • Unclear how statistical analysis of PPV or NPV  of a combination of localisations can be calculated

Minor comments

P2 ‘Transvaginal sonography is the first-line imaging technique to diagnose endometriosis, but magnetic resonance imaging (MRI) is more accurate to stage the extent of lesions, 64 especially for deep pelvic endometriosis (DPE) [3].’   This is not what ref 3 says in the text  ‘although evidence suggests that magnetic resonance imaging (MRI) is superior to ultrasound’ and in the conclusion

L75 ‘underlying’ -> underlining ?

Author Response

Reviewer 2

1

The major problems are that deep endometriosis and predictive values are poorly defined

Thanks you for this relevant comment.

However, we define the deep endometriosis according to BAZOT et al., definition based on previous reports on imaging (doi: 10.1148/radiol.2322030762;doi:10.1093/humrep/des211).

We strictly used the same consensual criteria to characterize and describe endometriosis and deep endometriosis in imaging.

In addition, we used data from surgery and pathological analysis to perform the correlation between pre-operative imagine and outcomes.

Therefore, to improve the quality of the manuscript we suggest to add the following sentence in the main text:

First in materials and methods section in the surgical and pathological finding :

“Deep pelvic endometriosis, also called deep infiltrating endometriosis, is defined as infiltration of the implant of endometriosis under the surface of the peritoneum as previously described[doi:10.1148/radiol.2322030762]. “

In the MRI evaluation we add the following sentence

“Briefly, as shown in Table 2, six Endo-stages MRI (0 to V) were created according to the anatomical location, extent of endometriotic lesions and ESHRE guidelines [doi:10.1093/hropen/hoaa002]. “

2

The MRI classification is a mixture of volume (stage 0 and I) extent (II  III )  and localisation (IV) of deep endo whereas VI is a poorly defined group including the diaphragm, the ileum, the appendix and the caecum.

Thank you for this very relevant comments which refers to the categorization and classification of lesion. We used this classification to help the physicians preoperatively.

We opted for stage 1 to stage 4 to be very descriptive due to their high prevalence in contrast to atypical or mixt lesion.

The stage VI is a specific entity requiring a pluridisciplinary approach.

The clinical value of this stages VI is to help the radiologist to refers the patients  directly to advances expert centers.

3

Since all women were operated on, it would be nice to start with the accuracy of estimated dimensions/presence/extend by MRI in comparison with surgery for each localization

This is a relevant issue but not the aim on this current publication.

The issue of the correlation between imaging and surgery for the  dimensions/presence/extend was previously stated by several authors.

In this setting, we also previously demonstrated for example that MRI is associated with underestimation of surgical finding of echo endoscopy (DOI: 10.1016/j.fertnstert.2008.09.005)

4

The area under the curve is a poor estimation of positive and negative predictive values.

We understand the comments.

In clinical practice the major issue face to a patient is to be able to predict or to inform and not to jeopardize the per and post-operative outcomes.

As an expert center our policy is to improve this information based on our experience, our pre-operative assessment and knowledge from published data.

We previously improved this aspect by numerus publications (doi: 10.1016/j.jmig.2010.08.692, DOI: 10.1016/j.jmig.2020.08.015, DOI: 10.1016/j.jmig.2019.07.011)

In this setting, no previous publications has stated the link between the descriptive anatomical location of endometriosis and the outcomes.

The philosophy of the present work is similarly to oncological gynecological cancer with the FIGO cancer and survival.

To help physician to improve the information quality for their patient.

5

The area under the curve is a poor estimation of positive and negative predictive values.

Thank you for this comment.

The PPV and NPV is relevant to estimate the accuracy of a diagnostic test.

In the current papiers the AUC was used as its ability to estimate the accuracy of our MRI classification to correctly identify and classify complication  (as previously reported in this article DOI: 10.1038/bjc.2015.35.) and not as diagnostic test.

6

In order to conclude that MRI is useful, an estimate of precision and confidence limits would be needed. 

Thank you for this comment.

We understand the comments and add the following information in the manuscript and Figure 7.

In the abstract we added the IC :

“On multivariate analysis, the Endo-StageMRI system of triage was strongly predictive of surgical complications and achieved a higher accuracy than the revised American Society for Reproductive Medicine classification [AUC: 0.78 (95% CI, 0.76-0.80) vs. 0.61 (95% CI, 0.58-0.64)].”

We also added all the IC in the result and the comment in the main text.

7

Unclear how statistical analysis of PPV or NPV  of a combination of localizations can be calculated

Thank you for this comment.

It is not necessary to estimate those value since the correlation is not between imaging and outcomes but between the classification and outcomes.

We estimate the value of the MRI classification as an entity and not the value of MRI finding.

8

P2 ‘Transvaginal sonography is the first-line imaging technique to diagnose endometriosis, but magnetic resonance imaging (MRI) is more accurate to stage the extent of lesions, 64 especially for deep pelvic endometriosis (DPE) [3].’   This is not what ref 3 says in the text  ‘although evidence suggests that magnetic resonance imaging (MRI) is superior to ultrasound’ and in the conclusion

Thank you for this comment.

We changed the reference page 2 in the main manuscript.

9

L75 ‘underlying’ -> underlining ?

Thank you for this comment.

It is indeed underlining. We made the modification L75 in the main manuscript

Reviewer 3 Report

Thank you very much for the invitation to review of the manuscript. It a great pleasure for me.

The purpose of study of Bazot et al. was to create a new imaging classification of endometriosis coined EndoStage MRI classification. That is a good paper, especially that there is no a good enough classification of the disease. However I have a few questions and comments:

  1. The abstract should be organized with a clear purpose.
  2. Do you take adhesion into account in its classification?
  3. Was there any correlation to the severity of the pain?
  4. I have a feeling that it does not cover all publications in the field, e.g. Foti et al., 2018, Celli et al., 2021 or Tong 2020.
  5. I find this classification not very intuitive, as 0 would mean no endometriosis. Why you decided to use that type of numbering?

Author Response

Reviewer 3

1

Thank you very much for the invitation to review of the manuscript. It a great pleasure for me.

The purpose of study of Bazot et al. was to create a new imaging classification of endometriosis coined EndoStage MRI classification. That is a good paper, especially that there is no a good enough classification of the disease. However I have a few questions and comments:

Thank you for this comment.

2

The abstract should be organized with a clear purpose.

Thank you for this comment, we reorganized the abstract such as (page 2): “Introduction: Transvaginal sonography is the first-line imaging technique to diagnose endometriosis, but magnetic resonance imaging is more accurate to stage the extent of lesions, especially for deep pelvic endometriosis.  The revised American Society for Reproductive Medicine and Enzian classifications are commonly used to stage the extent of endometriosis however, a review underlined their weaknesses in terms of complexity, lack of clinical reproducibility and low correlation with surgical complications and fertility outcomes. Thus, to this day, in clinical practice, there is a lack of consensual, standardized or common nomenclature to stage the extent of endometriosis, posing a worldwide challenge.

Objectives: The aims of our study were to: (i) develop a new classification (entitled Endo-Stage MRI) based on patterns of endometriosis as observed at magnetic resonance imaging; (ii) compare results with those of the rASRM classification; (iii) estimate the Endo-StageMRI accuracy to predict rate of surgical complications; and (iv) propose an Endo-StageMRI system of triage (low, intermediate, high) that correlates with the risk of surgical complications.

The goal is to improve the effectiveness of care pathways and allow for the planning of a multidisciplinary approach when necessary. “

3

Do you take adhesion into account in its classification?

Thank you for this specific and relevant issue and this answer is not yet.

We totally agree with the comment and we hope to be able to consider for the next version in the validation study the relevancy of this comments.

This idea reflects the experience of the reviewers since endometriosis is about lesion extent and lot of adhesion per operatively.

From an imaging point of view the criteria and the reproducibility of the criteria for adhesion description are still scarce.

4

Was there any correlation to the severity of the pain?

Thank you for this comment.

This is the major next issue of the following classification

We totally agree that issue is fundamental in the future to develop and adopt the following classification.

We are still working prospectively on this issue.

We propose to add a sentence in the limits sections to underline the high relevancy of those comments.

“Second, intra-and inter-observer variability was not evaluated and there is a lack of considerations of adhesions lesions in the different stage. In addition, the ENDO-stage MRI need to be evaluated from a clinical point of view with a specific correlation with symptom and quality of life. The following classification has been developed to improve description, classification and triage based on MRI findings. It would be interesting in the future to assess the classification value based on ultrasound and computed tomography especially for most advanced endometriosis lesion. Finally, our classification was created through a retrospective analysis of cases and not in a prospective fashion. So, a multicentric prospective is required to validate the potential value of such classification.”

5

I have a feeling that it does not cover all publications in the field, e.g. Foti et al., 2018, Celli et al., 2021 or Tong 2020.

Thank you for this comment.

We added the first publication after this sentence: However, the RCOG surgical score does not detail all the intraoperative complexities that may be encountered during surgery and is mainly intended to determine the level of expertise of surgeons [Foti et al].

We added the second and the third publication in the introduction in this sentence: “Transvaginal sonography is the first-line imaging technique to diagnose endometriosis, but magnetic resonance imaging (MRI) is more accurate to stage the extent of lesions, especially for deep pelvic endometriosis (DPE) [celli et al]

6

I find this classification not very intuitive, as 0 would mean no endometriosis. Why you decided to use that type of numbering?

We understand the comments. Stage 0 Endo-Stage MRI classification shows the presence of an endometrial cyst (E).

We use this numbering to be in line with all figo classification for gyunecolococal cancers oncological (stage I,II, III, IV).

Reviewer 4 Report

I would like to congratulate the authors for the well design research. MRI is one of non-invasive methods with high sensitivity for the diagnosis of endometriosis. This research is coming to add more information on how to use MRI mote effectively. It would be interesting if authors could include their opinion on how MRI compares to the other two alternatives, ultrasound and computed tomography.

Author Response

Reviewer 4

1

I would like to congratulate the authors for the well design research. MRI is one of non-invasive methods with high sensitivity for the diagnosis of endometriosis. This research is coming to add more information on how to use MRI more effectively.

Thank you for this comment.

2

It would be interesting if authors could include their opinion on how MRI compares to the other two alternatives, ultrasound and computed tomography.

We understand the comments.

Thank you for this very difficult issue. Concerning the ultrasound your idea is excellent and to be in line with this idea we will develop a study to estimate the value.

Concerning the tomography, in French guidelines and most specifically in our tertiary expert center we do not use tomography in the endometriosis assessment. However, maybe experienced teams on the subjects may lead to develop the idea.

We proposed to add the following sentence in the limits section.

The following classification has been developed to improve description, classification and triage based on MRI findings. It would be interesting in the future to asses the classificatio value based on ultrasound and computed tomography especially for most advanced endometriosis lesion.

Round 2

Reviewer 2 Report

The manuscript has not changed fundamentally. 

It remains unclear to me what benefit this classification will have for the clinician, in comparison with other existing classifications.

Please provide data, not opinions

Author Response

Reviewer 2

The manuscript has not changed fundamentally. 

It remains unclear to me what benefit this classification will have for the clinician, in comparison with other existing classifications.

Please provide data, not opinions

The existing classifications such as RASM, ENZIAN and DPEI have been previously developed to describe endometriosis lesion, extent and location. However to our knowledge  and after re reading each princeps manuscript no one has been developed to predict (i) surgical outcome in accordance to the anatomical description (ii) to help the triage of patients in surgical risk (low intermediate and high) since performance of the MRI imaging and at least (iii) to adopt a nomenclature similar to oncological filed with the FIGO stage classification known to be accurate and correlated to the prognosis.

With the current classifications we hypothesized the value of anatomical classification to promote the description, the triage and improvement of patients care pathways.

CLASSIFICATION

METHOD OF DEVELOPPEMENT (empirical or anatomical)

CLASSIFICATION ACCURACY

CORRELATION WITH SURGICAL OUTCOME

CORRELATION WITH SYMPTOMA

TRIAGE

CARE PATWAYS

ENZIAN
Tuttlies F, Keckstein J, Ulrich U, Possover M, Schweppe KW, Wustlich M, et al. ENZIAN-score, a classification of deep infiltrating endometriosis. Zentralbl Gynakol. 2005;127:275–81.

Empirical

Yes

No

No

No

No

dPEI
Thomassin-Naggara I, Lamrabet S, Crestani A, Bekhouche A, Wahab CA, Kermarrec E, Touboul C, Daraï E. Magnetic resonance imaging classification of deep pelvic endometriosis: description and impact on surgical management. Hum Reprod. 2020 Jul 1;35(7):1589-1600. doi: 10.1093/humrep/deaa103. PMID: 32619220.

Empirical

Yes

Yes

No

No

No

rASRM
Revised American Society for Reproductive Medicine classification of endometriosis: 1996. Fertil Steril. 1997 May;67(5):817-21. doi: 10.1016/s0015-0282(97)81391-x. PMID: 9130884.

Emprical

Yes

No

No

No

No

ENDO-STAGE MRI

Anatomical

Yes

Yes

Yes

Yes

Yes

In the result, we demonstrate “The respective AUC of Endo-StageMRI and r-ASRM classification for predicting overall complications rates, grade III-IV CDC and self-catheterization >1 month are reported in Figure 7. This indicates that Endo-StageMRI classification provides a higher accuracy than ASRM [AUC: 0.78 (95% CI, 0.76-0.80) vs. 0.61 (95% CI, 0.58-0.64)].

Round 3

Reviewer 2 Report

Concerns the manuscript “ ENDO_STAGE Magnetic Resonance Imaging: Classification to Screen Endometriosis

Specific comments

Non-accurate writing up to be misleading and selective use of references

  • Absence of line numbers to facilitate reviewing
  • Intro par 2 ‘but magnetic resonance imaging (MRI) is more accurate to stage the extent of lesions, especially for deep pelvic endometriosis’ superficial endometriosis is the most frequent and cannot be diagnosed by imaging. Even for deep endo the superiority of MRI is not that obvious except for sigmoid endometriosis.
  • Intro ‘However, a review underlined their weaknesses in terms of complexity, lack of clinical reproducibility and low correlation with surgical complications and fertility outcomes [8]’ Many articles discussed classifications underlining their lack of validation. Prediction of complications was not investigated in ref 8.
  • Today, 2022 for a new classification the reader would expect in the introduction what is the value and limitations of the existing surgical classifications, to understand the predictive value of imaging of endometriosis and why a pre-operative prediction would be useful.
  • M&M should mention whether MRI results were used as an indication for surgery
  • M&M: ‘surgical procedures previously published [15–20].’ Is mainly self-referencing while for the reader it is not clear which surgery was performed.
  • Deep endometriosis seems to be defined by MRI: ref 13 which is new
  • ‘All surgical, pathological, and outcomes findings documented in the official medical files…..’ what is all, limit to what was done.
  • ‘with more than 25 years of experience in MRI of endometriosis and a skilled surgeon (SB) with 5 years of surgical experience in endometriosis’ reads as self-promotion without helping the manuscript.
  • ‘Endo-Stage MRI classification’  and table 2 . This seems a surgical classification, not an MRI classification before surgery. Using this, to conclude is an MRI classification is misleading. That it predicts surgical difficulty is a self-fulfilling prophecy.
  • A correlation does not estimate the accuracy of classification. AUC  is not very informative for predictive values

Conclusion: the whole manuscript could be summarised as ‘bigger lesions of deep endometriosis affecting ureter and bowel’ cause more problems.  The added value of  MRI cannot be concluded from these data